# Serelaxin Alleviates Fibrosis in Thyroid-Associated Ophthalmopathy via the Notch Pathway

**DOI:** 10.3390/ijms24098356

**Published:** 2023-05-06

**Authors:** Anqi Sun, Huijing Ye, Zhihui Xu, Jingqiao Chen, Wei Xiao, Te Zhang, Xiaotong Sha, Shaowei Bi, Tianyi Zhou, Huasheng Yang

**Affiliations:** State Key Laboratory of Ophthalmology, Zhongshan Ophthalmic Center, Guangdong Provincial Key Laboratory of Ophthalmology and Visual Science, Sun Yat-sen University, Guangzhou 510060, China

**Keywords:** thyroid-associated ophthalmopathy, fibrosis, serelaxin, RXFP1, Notch signaling pathway

## Abstract

Fibrosis is the late stage of thyroid-associated ophthalmopathy (TAO), resulting in serious complications. Effective therapeutic drugs are still lacking. We aimed to explore the mechanism of TAO fibrosis and to find a targeted drug. High-throughput RNA sequencing was performed on orbital connective tissues from twelve patients with TAO and six healthy controls. Protein–protein interaction (PPI) network was constructed using the Search Tool for the Retrieval of Interacting Genes (STRING) database and we identified the hub gene by Cytoscape software. Additionally, the RNA sequencing results were validated by quantitative real-time polymerase chain reaction (qRT–PCR). Bioinformatic prediction identified the functions of differentially expressed genes (DEGs). Further orbital connective tissue and serum samples of the TAO and control groups were collected for subsequent experiments. Histologic staining, Western blotting (WB), qRT–PCR, enzyme-linked immunosorbent assays (ELISAs), gene overexpression through lentiviral infection or silencing gene by short interfering RNA (siRNA) were performed. We found that the relaxin signaling pathway is an important regulatory pathway in TAO fibrosis pathogenesis. Serelaxin exerts antifibrotic and anti-inflammatory effects in TAO. Furthermore, the downstream Notch pathway was activated by serelaxin and was essential to the antifibrotic effect of serelaxin in TAO. The antifibrotic effect of serelaxin is dependent on RXFP1.

## 1. Introduction

Thyroid-associated ophthalmopathy (TAO) has the highest incidence among adult orbital diseases [1]. As an autoimmune disease, TAO is characterized by sensitive T cells and antibodies against autoantigens, such as thyrotropin receptor (TSHR) and insulin-like growth factor-1 receptor (IGF-1R) [2,3]. In the initial stage, immune cells infiltrate the orbital connective tissues, and orbital fibroblasts (OFs) are activated to proliferate, synthesizing glycosaminoglycan (GAG) and secreting chemoattractants to recruit more immune cells [4]. Orbital soft tissue remodeling and fibrosis dominate the final stage [5]. As the underlying mechanism of TAO is unclear, targeted treatments are lacking. Glucocorticoids are the mainstay treatment for patients with active-phase moderate to severe TAO, and their efficacy is approximately 83% [6,7]. However, there are no effective drugs widely accepted for treating fibrosis in the late stage of TAO.

Serelaxin is the recombinant form of the human hormone relaxin-2 (RLN2). RLN2 can be produced by reproductive and nonreproductive organs, considered to be one of the endogenous hormones with the strongest antifibrotic effect [8,9,10,11] This peptide hormone was confirmed to exert antifibrotic effects in vitro and in vivo via its specific receptor RXFP1 in multiple-organ fibrosis [12], showing good promise for clinical application [13]. Furthermore, our previous research revealed that the relaxin signaling pathway is crucial in TAO clinical subtyping and fibrosis is one of the features of type II TAO [14]. However, the antifibrotic effect of serelaxin on TAO still lacks relevant research. 

In this study, high-throughput RNA sequencing (RNA-seq) was performed on TAO orbital connective tissues and healthy control samples. Differentially expressed gene (DEG) patterns were revealed, and the relaxin signaling pathway was identified as an important regulatory axis in the pathogenesis of TAO. Furthermore, the antifibrotic effect and mechanism of serelaxin were explored in a TAO in vitro model. The effect of RXFP1 expression on serelaxin treatment sensitivity was also investigated. The experimental results indicated that serelaxin may be a candidate drug for the treatment of fibrosis in TAO.

## 2. Results

### 2.1. mRNA Expression Profile and Functional Enrichment Analysis of DEGs

The selected orbital connective tissues for high-throughput transcriptome sequencing were those with obvious fibrosis (Appendix A). A total of 15,803 mRNAs were predicted from the RNA-seq data. A total of 550 mRNAs were significantly differentially expressed between the two groups (FC ≥ 2.0 and P-adj < 0.05), with 314 mRNAs upregulated and 236 mRNAs downregulated in the TAO group compared with the control group. To clarify the potential biological function and signaling pathways of DEGs, we used KEGG analyses to determine the functions of these genes. A total of 35 signaling pathways were revealed by KEGG analysis, with the relaxin signaling pathway included (Figure 1).

### 2.2. Validation of DEGs in the Relaxin Signaling Pathway

A PPI network was constructed with DEGs, and the hub genes were identified. COL1A1, MMP2, and FOS were among the top 30 hub genes, while GNG4 and CREB5 were closely related to the activation of the relaxin pathway. Thus, we chose these five genes contained in relaxin pathway to verify the RNA-seq outcomes using orbital connective tissue. The qRT–PCR results were consistent with the RNA-seq results. We speculated that the relaxin signaling pathway is regulated by the differential expression of these major genes which leads to the occurrence and development of TAO, especially fibrosis (Figure 2).

### 2.3. Relaxin Signaling Pathway Expression Level in Human Orbital Connective Tissue

As the activation of the relaxin signaling pathway starts from the binding of RLN2 to its cognate receptor RXFP1 [15], we detected the expression levels of these signaling pathway markers according to previous research [16]. RLN2 and RXFP1 expression was detected in both TAO and control samples by IHC. The expression level of TAO orbital connective tissue was significantly higher than that in the control groups. Furthermore, within the area of fibrosis, the relaxin pathway was significantly activated (Figure 3).

### 2.4. RLN2 Expression in Serum and Supernatants

The serum RLN2 in the TAO group was higher than that in the control group. Moreover, TGF-β1 increased the secretion of RLN2 in the supernatant. These results suggested that RLN2 participates in the process of fibrosis in TAO and is derived from circulation and autocrine signaling (Figure 4).

### 2.5. Serelaxin Alleviates Fibrosis in OFs of TAO

The cell viability did not decrease when the serelaxin concentration was increased to 2000 ng/mL (Figure 5A). A dose-dependent experiment was subsequently used to identify the optimum concentration. The antifibrotic effect improved with concentration at first, but at a concentration of 1000 ng/mL, the antifibrotic effect no longer increased (Figure 5B). Thus, we chose 1000 ng/mL as the appropriate concentration for the subsequent experiments. TGF-β1 was used to construct an in vitro fibrosis model. We found that serelaxin could decrease the expression levels of α-SMA, COL1A1, FN1, and TIMP1 that were increased by TGF-β1 stimulation (Figure 5C–J). To investigate the underlying mechanism by which serelaxin inhibits fibrosis induced by TGF-β1, we subsequently screened several classical pathways, such as P38/MAPK, STAT3, ERK, and the Notch signaling pathway, using WB. However, the phosphorylation of P38, STAT3, and ERK were not changed after RLN2 stimulation. As shown by Western blotting, the NICD and Jagged-1 expression levels were significantly decreased in the TGF-β1-simulated group compared with the control group. However, this effect was weakened by serelaxin (Figure 6A–C). Additionally, serelaxin inhibited the TGF--β1/smad2 pathway (Figure 6D,E). To further investigate the role of the Notch pathway in the serelaxin antifibrotic process, we used DAPT, a pharmacological inhibitor of Notch signaling. We found that the NICD expression level was decreased by DAPT. Moreover, inhibiting the Notch pathway weakened the antifibrotic effect of serelaxin (Figure 6F–H). Overall, we speculated that serelaxin alleviated fibrosis via the Notch pathway in TAO.

### 2.6. The Antifibrotic Effect of Serelaxin Is Dependent on RXFP1

To investigate the role of the RLN2 receptor (RXFP1), we first knocked down the gene RXFP1 by siRNA. With the same concentration of serelaxin, after RXFP1 knockdown, the inhibition of fibroblast differentiation to myofibroblasts was abolished (Figure 7A–E). In contrast, with overexpression of RXFP1 by lentiviral transfection, the RXFP1 group showed decreased α-SMA expression even with a halved concentration of serelaxin (Figure 7F–J). Thus, the antifibrotic effect of serelaxin is dependent on its receptor RXFP1 in TAO. Expression of receptors can improve OFs sensitivity to serelaxin.

### 2.7. TGF-β1 Regulates RXFP1 Expression in OFs

TGF-β1 is a major cytokine in the fibrotic process. We used different concentrations of TGF-β1 to stimulate OFs of TAO. Interestingly, the expression level of RXFP1 first increased and subsequently decreased (Figure 8A–C).

### 2.8. Serelaxin Alleviates Inflammation in the OFs of TAO

Fibrosis can be considered the late stage of TAO, repairing tissue damage caused by chronic inflammation. Therefore, we researched the anti-inflammatory effect of serelaxin. Cells were pretreated with serelaxin (1000 ng/mL) for 24 h, and then, IL-1β (1 ng/mL) was added to establish an inflammation model. Serelaxin demonstrated a significant anti-inflammatory effect in TAO OFs, as shown by RT–qPCR (Figure 9A–D). Next, we detected the IL-6 and IL-8 expression levels in the supernatant by ELISAs. Serelaxin blocked inflammatory changes induced by IL-1β (Figure 9E,F). However, the anti-inflammatory effect of serelaxin was much less than dexamethasone (Appendix A).

## 3. Discussion

TAO is the most common disorder in orbital disease and can develop into sight-threatening ocular disease [17]. The remodeling of orbital soft tissue leads to disfigurement, strabismus, and even vision loss and a decreased quality of life [18]. Injury and inflammation, among various causes, are stimulators of fibrosis [19]. Fibrosis and tissue remodeling dominate the final stage of TAO. However, therapeutic strategies for fibrosis in TAO are lacking.

In this study, high-throughput RNA-seq was performed on twelve TAO and six control orbital connective tissues to reveal DEGs to determine the underlying mechanism of TAO, and quarry the effective treatment methods. All twelve TAO patients were in an inactive (stable) phase; the average duration time of TAO was 23.83 ± 8.21 months, indicating fibrosis dominating the stage according to Rundle’s curve [20]. Through functional enrichment analysis of DEGs, the relaxin signaling pathway was found to be a vital regulatory axis and the five signaling pathways in front of it have already been investigated [21,22]. According to previous numerous research studies, the relaxin signaling pathway is considered a classical antifibrotic pathway [23], with good prospects for clinical application [24]. Thus, we concentrated on the relaxin signaling pathway. 

The hub and significant genes involved in this pathway were validated. COL1A1 is an important component of the ECM, and MMP2 has hydrolytic activity on ECM proteins [25,26]. COL1A1 and MMP2 play an important role in the development of TAO fibrosis [27]. FOS has been proven to be a regulator of cell proliferation, differentiation and apoptosis [28]. Tissues obtained from TAO exhibited different mRNA levels of FOS [29]. CREB5 is considered an important gene in fibrosis. A recent study showed that CREB lncRNA targeted CREB5 to regulate FN1 in renal fibrosis [30]. GNG4 is a G protein subunit that regulates various transmembrane signaling pathways [31]. GNG4 inhibits cell proliferation and cell migration and promotes cell apoptosis [32]. Above all, these genes are vital in the progression and development of TAO. Thus, we speculated that the differential expression of COL1A1, MMP2, FOS, GNG4, and CREB5 abnormally regulated the relaxin signaling pathway to affect disease progression. 

To further validate the KEGG analyses, we performed IF and IHC to evaluate the protein expression level of the relaxin signaling pathway in TAO, and the marker of this pathway was detected. The data showed that RLN2 and RXFP1 levels were significantly higher in the TAO group than in the control group and relevant to fibrosis. This finding indicated that the relaxin signaling pathway was activated in TAO. Additionally, we discovered that the serum and supernatant concentrations of RLN2 were apparently higher in the TAO group than in the control group. This finding suggested that RLN2 was derived from circulation and autocrine signaling, consistent with recent studies [33]. For example, nonreproductive organs such as hyper- and neoplastic human thyrocytes could be a source and target of relaxin [34].

Serelaxin is the recombinant form of human hormone RLN2. We chose serelaxin to explore the anti-fibrotic effect of the relaxin pathway in TAO. Firstly, cell viability was not significantly affected by serelaxin. A dose-dependent experiment was subsequently performed, and 1000 ng/mL was the optimal concentration, consistent with a previous study [35]. In the pathophysiology of TAO, OFs act as both target and effector cells [36]. In the development of fibrosis, OFs differentiate into myofibroblasts expressing the hallmark gene a-smooth muscle actin (a-SMA) and synthesize excessive ECM components, such as fibronectin and collagen [37,38]. TGF-β1 is a well-known cytokine that induces fibroblast transformation to myofibroblasts. In our study, we found that the elevated expression levels of a-SMA, COL1A1 and FN1 induced by stimulation with TGF-β1 could be inhibited by serelaxin, as shown by PCR and WB. Tissue inhibitors of metalloproteinase (TIMP) are inhibitors of matrix metalloproteinase, and the balance between TIMP and MMP affects the amount of ECM [39]. The TIMP1 expression level was increased after stimulation with TGF-β1. However, serelaxin counteracted this effect. We concluded that serelaxin had a therapeutic effect on alleviating fibrosis in TAO. To clarify the underlying mechanism by which serelaxin inhibited fibrosis in TAO, we concentrated on the canonical Notch signaling pathway. The Notch pathway is a highly conserved signaling pathway generated by adjacent cell interactions [40]. After three cleavage reactions in the trans-Golgi network, A Disintegrin and Metalloprotease (ADAM) and the γ-secretase complex NICD were produced [41]. Previous studies have revealed that the Notch signaling pathway is involved in fibrosis, including that of multiple organs [42,43,44,45,46]. However, the Notch pathway may induce opposite regulatory effects depending on the tissue context and disease states [47]. Regardless of this, the Notch signaling pathway is vital to fibrosis. Earlier studies demonstrated that serelaxin prevented TGF-β1-induced suppression of the Notch signaling pathway to alleviate fibrosis [48,49,50]. Our research showed the same results. The reduction in NICD and Jagged-1 induced by TGF-β1 was reversed by serelaxin. To better understand the interaction of serelaxin and the Notch pathway in TAO fibrosis, we added DAPT to the cell culture medium. We observed that DAPT inhibited the Notch pathway and decreased the production of NICD. Moreover, through this suppression, the effect of serelaxin on alleviating fibrosis was also prevented. In summary, serelaxin exerts antifibrotic therapeutic effects via the Notch pathway in OFs.

RLN2 is the main circulating and stored relaxin form in humans, and RXFP1 is its primary receptor. We initially found that the expression of RXFP1 in patients with TAO was higher than that in the controls. To explore the effect of RXFP1 on the antifibrotic effect of RLN2, we knocked down the RXFP1 gene by siRNA. The results indicated that although the concentration of serelaxin was not changed, the ability to inhibit fibroblast differentiation to myofibroblasts was weakened compared to that in the control group. In contrast, when RXFP1 was overexpressed by the lentiviral vector, although the concentration was halved, serelaxin could also prevent the increased expression of α-SMA induced by TGF-β1 compared to that in the GFP group. Thus, we speculated that the antifibrotic effect of serelaxin is dependent on RXFP1, and higher expression of RXFP1 may indicate enhanced sensitivity to serelaxin treatment. 

Subsequently, the relationship between TGF-β1 and RXFP1 expression was explored. In different organs, the expression of RXFP1 could decrease or increase because of heterogeneity of fibrosis [51,52]. Interestingly, RXFP1 expression increased at relatively low concentrations of TGF-β1 and then decreased with increasing concentrations in TAO. As inflammation can be a trigger for fibrosis, we tested the anti-inflammatory effect of serelaxin. IL-1β is a proinflammatory cytokine secreted by OFs in TAO [53]. Serelaxin could block inflammatory changes induced by IL-1β. During the inflammatory stage, the cytokines IL-6 and IL-8 are persistently produced [54]. Chemokines such as C-X-C motif chemokine ligand 1 (CXCL1), CXCL2, C-C motif chemokine ligand 2 (CCL2) and CCL8 recruit leukocytes from blood to local tissue [55,56]. In this study, we revealed that serelaxin mitigated the inflammatory response by decreasing inflammatory cytokines and chemokines. However, the anti-inflammatory effect of serelaxin was much less than dexamethasone. Previous research has demonstrated that the expression of TGF-β1 was positively correlated with CAS in TAO [57]. Based on the above viewpoints, we speculated that the combination with other anti-inflammation drugs such as glucocorticoids in active phase and serelaxin alone in inactive phase would induce a better anti-inflammatory and antifibrosis effect on TAO. 

In summary, we revealed that the relaxin signaling pathway is an essential regulatory axis in TAO pathology. Serelaxin may be a candidate drug for the treatment of fibrosis in TAO.

## 4. Materials and Methods

### 4.1. Patient Selection and Tissue Samples

We collected orbital connective tissues from 12 patients with TAO and 6 control subjects from Zhongshan Ophthalmic Center at Sun Yat-sen University for High-throughput RNA-sequencing. Another 6 TAO and 6 control orbital samples were collected for validation and other experiments (Table 1). Serum samples were from 22 TAO patients and 10 healthy volunteers (Table 2). The enrolled patients with TAO met the standards according to Bartley [58]. Patients who had received glucocorticoids or immunotherapy treatment in the last 3 months were excluded. Orbital connective tissues were collected during orbital decompression surgery. Adipose tissue from the control group was derived from enucleation surgery having been diagnosed with an intraocular tumor. All the patients understood the purpose of the investigation and agreed to cooperate with the study. This research was approved by the Institutional Review Board of Zhongshan Ophthalmic Center (2016KYPJ028).

### 4.2. RNA Sequencing

Total RNA was extracted from orbital connective tissue with a HiPure Total RNA Mini Kit (Magen, Guangzhou, China) according to the manufacturer’s protocol. Libraries were constructed using the Total RNA-seq (H/M/R) Library Prep Kit for Illumina^®^. Genes showing altered expression with *p* < 0.05 and more than 1.5-fold changes were considered differentially expressed.

### 4.3. Functional Enrichment Analysis

For the DEGs set and the functional genes of the corresponding species in the Kyoto Encyclopedia of Genes and Genomes (KEGG) database, the R software cluster Profiler was used to calculate the intersection of the two sets to obtain statistical significance and to determine the roles of the DEG sets.

### 4.4. Construction of a PPI Network and Identification of Hub Genes

The Search Tool for the Retrieval of Interacting Genes (STRING) database was used to construct a PPI network for DEGs between the TAO and control groups, and then, the top 30 hub genes were identified with CytoHubba, a plugin of Cytoscape software (version 3.8.2).

### 4.5. Quantitative Real-Time Polymerase Chain Reaction (qRT–PCR)

Total RNA of cells or orbital explants was extracted with a kit (ESscience, Shanghai, China). Then, RNA was used to synthesize cDNA with PrimeScript RT Master Mix (TaKaRa, Dalian, China). The relative expression levels of RNAs were detected by a Roche LightCycler 480 (Roche, Basel, Switzerland). TB Green Premix Ex Taq II (TaKaRa, Dalian, China) was used. The primer sequences were listed in Appendix A. GAPDH was selected as a reference gene.

### 4.6. Histologic Staining

Formalin-fixed, paraffin-embedded orbital samples were sectioned, deparaffinized and rehydrated. The sections were stained with primary antibodies against relaxin-2 (Abcam, Waltham, MA, USA) and RXFP1 (Bioss, Beijing, China) and secondary antibodies (Cell Signaling Technology, Boston, USA) for Immunohistochemistry. Masson trichrome staining (Wuhan, China, Servicebio) was used for evaluating tissue fibrosis.

### 4.7. Immunofluorescence (IF)

Tissue sample sections were deparaffinized, rehydrated, blocked, and then incubated with primary antibodies against relaxin-2 (Abcam, Waltham, USA) and RXFP1 (Bioss, Beijing, China) and probed with secondary antibodies (both from Nanoprobes, New York, NY, USA). Nuclei were stained with DAPI (Bioss, Beijing, China). Finally, the samples were sealed and mounted. The observation and imaging system used a confocal microscope (Carl Zeiss 880, Oberkochen, Germany).

### 4.8. Cell Culture

Fresh orbital connective tissue was removed from blood vessels, cut into small pieces, and then placed into Petri dishes with Dulbecco’s modified Eagle’s medium (DMEM) (Gibco, New York, NY, USA) with 10% fetal bovine serum (FBS) (Gibco), 1% streptomycin (100 mg/mL) and penicillin (100 mg/mL) (Gibco) for approximately 2 weeks at 37 °C. OFs were used from passages 3 to 6. Cells were treated with TGF-β1 (5 ng/mL) (R&D Systems, Minneapolis, MN, USA), IL-1β (1 ng/mL) (R&D), (rh)RLN2 (1000 ng/mL) (Peprotech, Cranbury, NJ, USA), DAPT(20μM) (Cell Signaling Technology), Dexamethasone (10 nM) (Sigma-Aldrich, Shanghai, China) in DMEM supplemented with 1% FBS. To synchronize the cell cycle, the cells were cultured in DMEM supplemented with 1% FBS before stimulating. For qRT–PCR, the incubation period was 24 h, and WB was 72 h.

### 4.9. Cell Proliferation/Toxicity Assay

Cell Counting Kit-8 (Dojindo, Shanghai, China) was used to assess cell viability at 24 h, 48 h, and 72 h with different concentrations of (rh)RLN2.

### 4.10. Western Blotting

Orbital tissue and OFs were lysed to extract protein with a protein extraction kit (KeyGEN, Nanjing, China). The concentration was measured using a BCA (Cwbiotech, Beijing, China). Protein lysates were electrophoresed, transferred to membranes, blocked and incubated with primary and secondary antibodies. Finally, images were obtained with a chemiluminescence imager (Tanon Science & Technology, Shanghai, China). All antibodies were purchased from Cell Signaling Technology.

### 4.11. Enzyme-Linked Immunosorbent Assay (ELISA)

According to the manufacturer’s protocol, the concentrations of IL-6 and IL-8 in the cell culture supernatant were quantified with an ELISA kit (R&D). The RLN2 concentration was detected with an ELISA kit (Animalunion Biotechnology, Shanghai, China). Absorbance was detected at 450 nm.

### 4.12. Overexpression or Silencing of RXFP1

A lentiviral vector was constructed encoding RXFP1 or green fluorescence protein (GFP) (Hanbio Biotechnology, Shanghai, China). Gene silencing used short interfering RNA (siRNA) targeting RXFP1 or nontargeting controls (siControl) (HanYi Bioscience, Guangzhou, China).

### 4.13. Statistical Analysis

GraphPad Prism (Prism 8.0.1, GraphPad Software, San Diego, CA, USA) was used to analyze the data and draw the figures. For a comparison of the quantitative data differences between the two groups, a t test was used. The descriptive data, such as rates and ratios, were analyzed with the χ2 test. A *p* value < 0.05 was considered statistically significant.

## Figures and Tables

**Figure 1 ijms-24-08356-f001:**
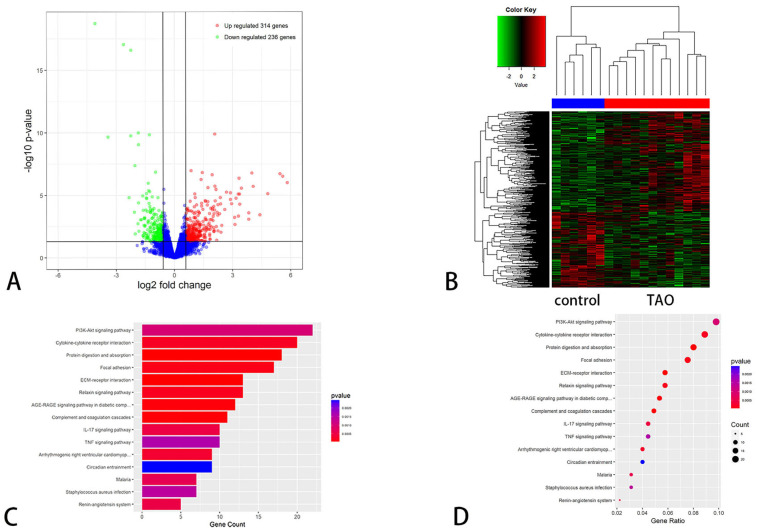
Results of RNA-seq in the TAO and control orbital connective tissue. (**A**) Volcano plot of mRNAs that were differentially expressed between the TAO and control groups. (**B**) Hierarchical cluster analysis of differentially expressed mRNAs. (**C**) Bar diagram of Kyoto Encyclopedia of Genes and Genomes (KEGG) pathway analysis of the TAO and control groups. (**D**) Bubble chart of the KEGG pathway analysis of the TAO and control groups.

**Figure 2 ijms-24-08356-f002:**
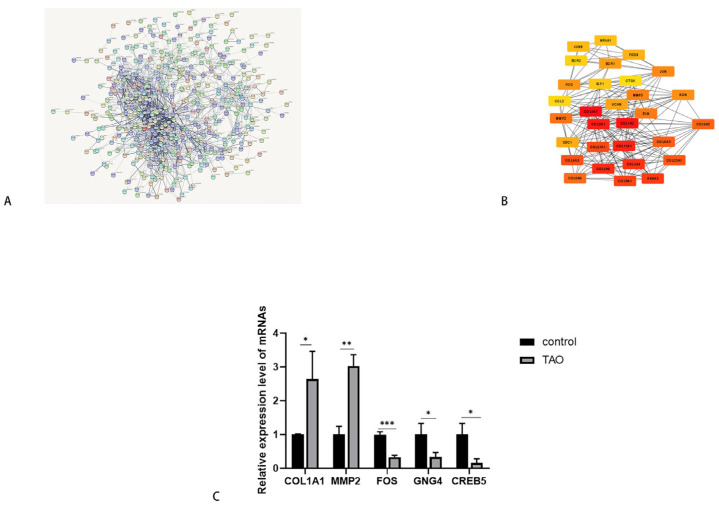
Top 30 hub genes identified in the PPI network and validation of selected DEGs. (**A**) PPI network construction for differentially expressed genes. (**B**) The top 30 hub genes. (**C**) Validation for selected DEGs by qRT–PCR using connective tissue (*n* = 3). * *p* < 0.05, ** *p* < 0.01, *** *p* < 0.001 compared with the control group. PPI: protein–protein interaction. DEG: differentially expressed gene.

**Figure 3 ijms-24-08356-f003:**
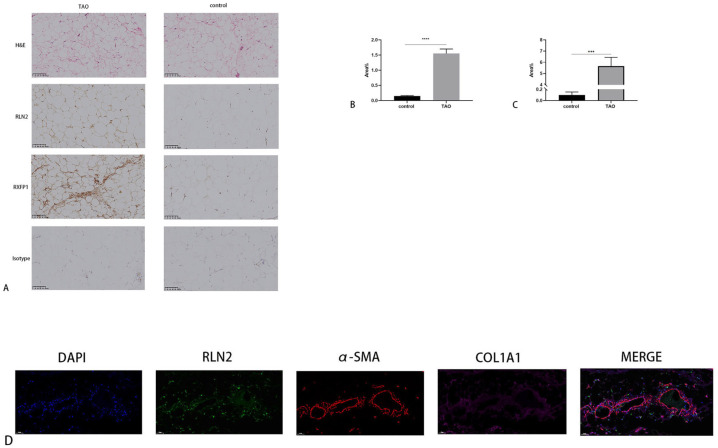
RLN2 and RXFP1 expression in human orbital connective tissue. (**A**) Immunohistochemistry (IHC) of RLN2 and RXFP1. Scale bars, 100 μm. (**B**,**C**) The positive area ratio of RLN2 and RXFP1 in a comparison of the TAO to control groups. (**D**) Immunofluorescence (IF) of RLN2, α-SMA and COL1A1 in orbital connective tissue from the TAO. Magnification, 20× (*n* = 3) *** *p* < 0.001, **** *p* < 0.0001, compared with the control group.

**Figure 4 ijms-24-08356-f004:**
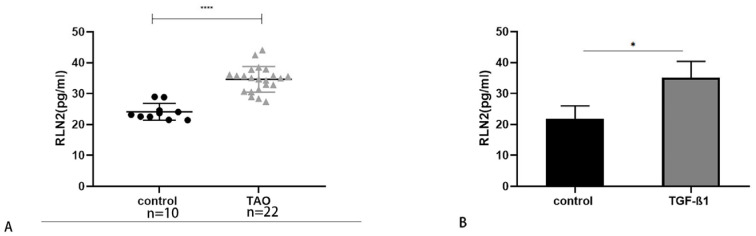
RLN2 expression in serum and supernatants. (**A**) The RLN2 expression level in serum of TAO patients (*n* = 22) compared with the controls by ELISA (*n* = 10). (**B**) The expression level of RLN2 after TGF-β1 stimulation and control in the orbital fibroblast supernatant by ELISA (*n* = 3). * *p* < 0.05, **** *p* < 0.0001, compared with the control group.

**Figure 5 ijms-24-08356-f005:**
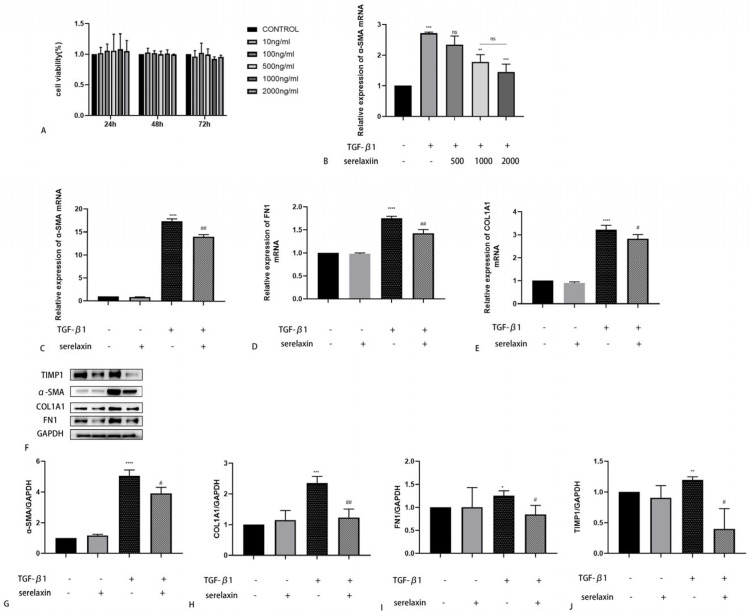
Serelaxin exerts an antifibrotic effect on TAO OFs. (**A**) Cell viability was assessed by CCK8 assays at different concentrations and time points. (**B**) The dose-dependent experiments of serelaxin. (**C**–**E**) The mRNA levels of fibrotic markers (α-SMA, FN1 and COL1A1) were measured. (**F**) The protein expression levels of the fibrotic markers were detected by WB. (**G**–**J**). The protein level was quantified and analyzed (*n* = 3). * *p* < 0.05, ** *p* < 0.01, *** *p* < 0.001, and **** *p* < 0.0001 compared with the controls; # *p* < 0.05 and ## *p* < 0.01 compared with TGF-β1; ns denotes no statistical significance between groups.

**Figure 6 ijms-24-08356-f006:**
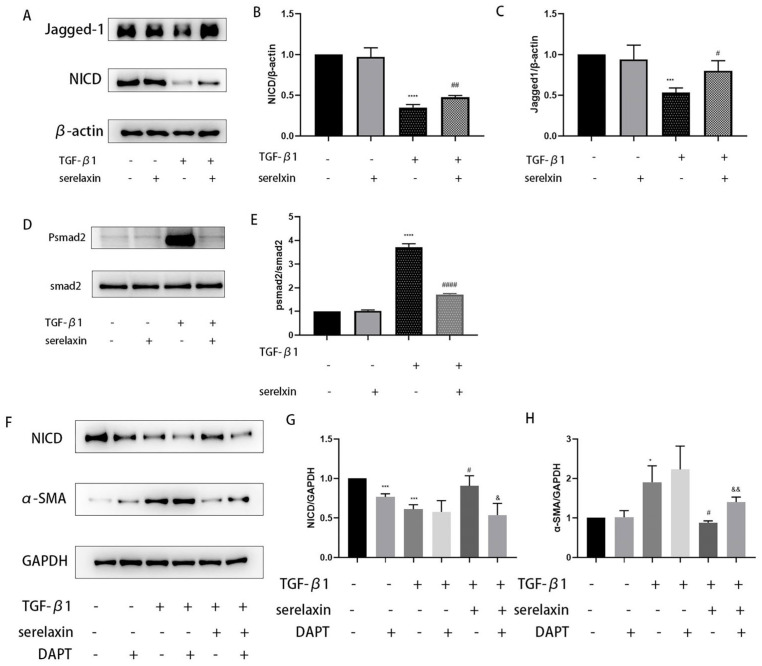
The serelaxin and Notch signaling pathways negatively regulate the TGF-β1-induced fibroblast-myofibroblast transition in TAO OFs. (**A**) Serelaxin weakened the inhibitory effect of the Notch signaling pathway induced by TGF-β1. (**B**,**C**) The protein level of NICD and Jagged-1 was quantified and analyzed. (**D**) Serelaxin inhibited TGF-β1/smad2 pathway. (**E**) The protein level of p-smad2. (**F**) Western blotting analysis of NICD and α-SMA expression in the absence or presence of DAPT and serelaxin. (**G**,**H**) The protein level of NICD and α-SMA was quantified and analyzed with or without DAPT and serelaxin (*n* = 3). * *p* < 0.05, *** *p* < 0.001 and **** *p* < 0.001 compared with the control; # *p* < 0.05, ## *p* < 0.01 and #### *p* < 0.0001 compared with TGF-β1; & *p* < 0.05 and && *p* < 0.01 represent TGF-β1 with serelaxin compared to TGF-β1 with serelaxin and DAPT.

**Figure 7 ijms-24-08356-f007:**
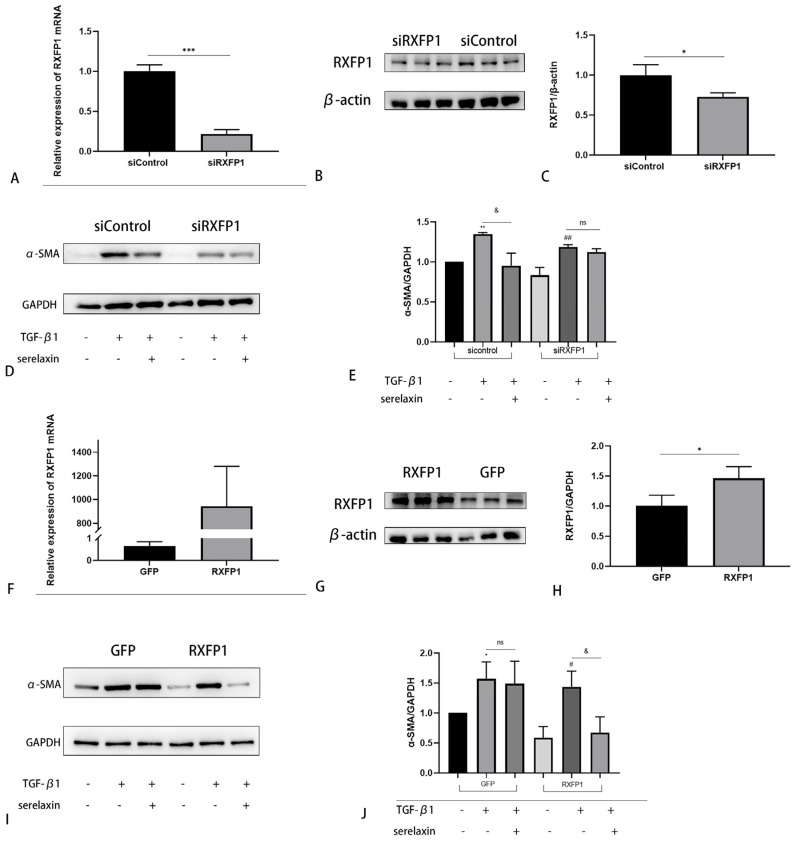
The sensitivity of OFs to serelaxin is dependent on RXFP1 expression. (**A**–**C**) The mRNA and protein levels of RXFP1 and the expression level were quantified and analyzed in the siControl and siRXFP1 groups. OFs were incubated with siControl or siRXFP1. (**D**,**E**) The protein level of α-SMA in the siRXFP1 group and the siControl group, with or without TGF-β1, and serelaxin and its expression level were quantified and analyzed. The concentration of serelaxin was the same in the two groups (*n* = 3). ** *p* < 0.01, *** *p* < 0.001 compared to the control in the siControl group. ## *p* < 0.01 compared to the control in the siRXFP1 group. & *p* < 0.05. (**F**–**H**) The mRNA and protein level of RXFP1 and the expression level was quantified and analyzed in the GFP and RXFP1 groups. (**I**,**J**) The protein level of α-SMA in the RXFP1 group and the GFP group with or without TGF-β1 and serelaxin was quantified and analyzed. The concentration of serelaxin was half of the normal level (*n* = 3). * *p* < 0.05 compared to the control in the GFP group. # *p* < 0.05 compared to the control in the RXFP1 group. & *p* < 0.05. siControl— nontargeting short interfering RNA (siRNA) oligonucleotides. siRXFP1—RXFP1-targeting siRNA oligonucleotides. GFP group—green florescent protein. RXFP1 group—RXFP1-expressing lentiviral particles.

**Figure 8 ijms-24-08356-f008:**
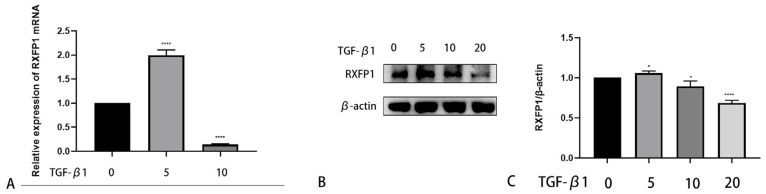
TGF-β1 regulates RXFP1 expression in human OFs. (**A**) The mRNA expression of RXFP1 with elevated concentrations of TGF-β1 (ng/mL). (**B**) Western blotting of the protein level of RXFP1 with different concentrations of TGF-β1 (ng/mL). (**C**) The protein level of RXFP1 was quantified and analyzed. *n* = 3. * *p* < 0.05, **** *p* < 0.0001 compared with the controls.

**Figure 9 ijms-24-08356-f009:**
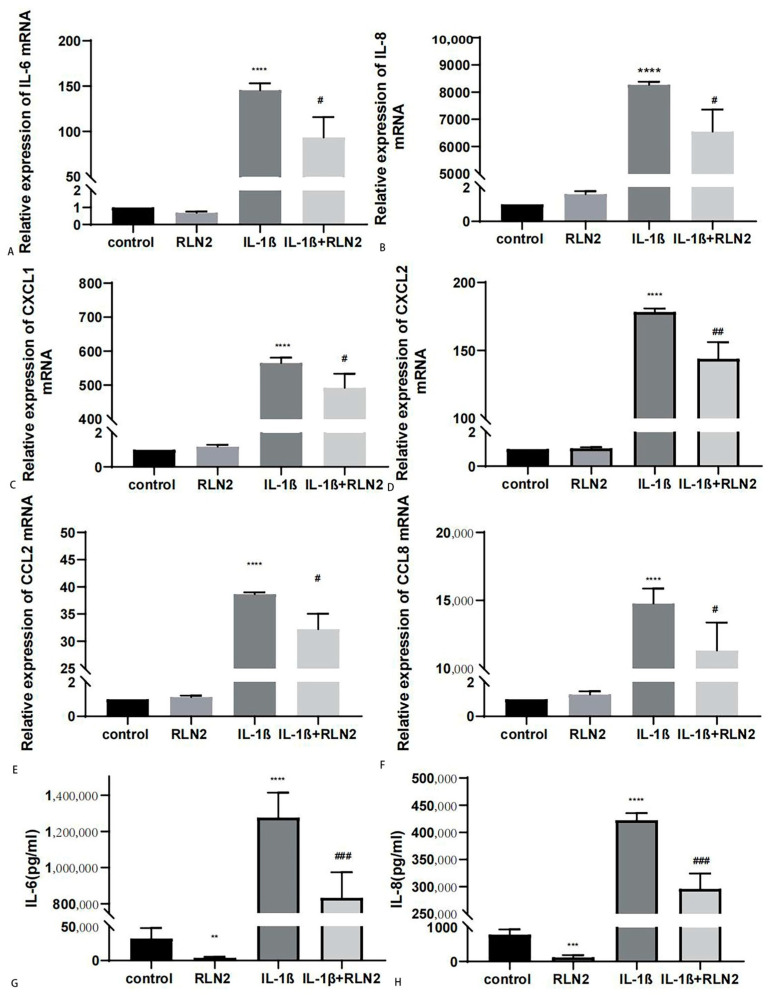
Serelaxin exerts an anti-inflammatory effect on TAO OFs. (**A**–**F**) The mRNA levels of inflammatory markers (IL-6, IL-8, CXCL1, CXCL2, CCL2, CCL8). *n* = 3. (**G**) The protein level of IL-6 detected by ELISA. *n* = 6. (**H**) The protein level of IL-8 detected by ELISA. *n* = 4. ** *p* < 0.01, *** *p* < 0.001, **** *p* < 0.0001 compared with the controls. # *p* < 0.05, ## *p* < 0.01 and ### *p* < 0.001 compared with IL-1β.

**Table 1 ijms-24-08356-t001:** Demographic data of orbital connective tissue samples used in this research.

Clinical Characteristics	RNA-Sequencing Stage			Experiments Stage		
	TAO (*n* = 12)	Control (*n* = 6)	*p*	TAO (*n* = 6)	Control (*n* = 6)	*p*
Age, y	41.5 ± 13.5	44.5 ± 12.8	0.657	47.2 ± 13.0	46.7 ± 13.5	0.949
Sex, M/F	4/8	3/3	0.494	4/2	3/3	0.558
CAS	1 (range: 0–2)	-	NA	1 (range: 0–2)	-	NA
NOSPECS score	4 (range: 3–6)	-	NA	4 (range: 3–6)	-	NA
TAO duration, mo	25 (IQR: 18–29.8; range: 12–36)	-	NA	27.5 (IQR: 22–34.3; range: 19–35)	-	NA

TAO, Thyroid-associated ophthalmopathy; M, Male; F, Female. CAS, Clinical Activity Score. mo, month; IQR, interquartile range. Age is reported as mean ± SD. CAS, NOSPECS score and duration of TAO are reported as median.

**Table 2 ijms-24-08356-t002:** Clinical characteristics data of serum subjects used in this study.

	TAO (*n* = 22)	Control (*n* = 10)	*p*
Sex (M/F)	13/9	6/4	0.961
Age (years)	47.68 ± 10.13	41.6 ± 12.23	0.1389
CAS	2 (range: 0–2)	-	NA
Duration of TAO (mo)	17.5 (IQR: 12–24; range: 12–38)	-	NA
Therapy History	Glucocorticoids: 7/22 Thyroid surgery: 3/22	-	NA
Thyroid Function	Euthyroid: 20/22 Hypothyroid: 2/22 Hyperthyroid: 0/22	-	NA

TAO, Thyroid-associated ophthalmopathy; M, Male; F, Female. CAS, Clinical Activity Score. mo, month; IQR, interquartile range. Age is reported as mean ± SD. CAS and Duration of TAO are reported as median.

## Data Availability

The datasets generated during and/or analyzed during the current study are not publicly available but are available from the corresponding author on reasonable request.

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
