# Peer review of "Serelaxin Alleviates Fibrosis in Thyroid-Associated Ophthalmopathy via the Notch Pathway"

_ijms, 2023, doi:10.3390/ijms24098356_

Round 1

Reviewer 1 Report

Interesting project to understand tissue fibrosis seen in thyroid eye disease in the more chronic phase.    Given human tissue was used, results are likely to be clinically significant. Clearly presented experiments to support conclusions about the mechanisms of fibrosis found in TAO and to investigate specific molecular pathways that could be targets for possible treatment/ prevention of tissue changes in TAO. 

Author Response

Thank you for your comments on the paper. 

Reviewer 2 Report

Although the number of cases is limited, we believe that it is a new finding and appropriate for publication.

Author Response

(The authors gave the same response as above.)

Reviewer 3 Report

The authors have revealed that the relaxin signaling pathway is an essential regulatory pathway in TAO pathology, and serelaxin may be a candidate drug for the treatment of fibrosis in TAO. The study is appreciated as a proposal for a possible novel treatment strategy. However, the following points need clarification:

Fibrosis is known to be dependent on TGF-beta signaling, especially the Smad signal. Will the authors investigate the crosstalk between relaxin signaling and Smad2/3?

While it is understood that many cell types are involved in TAO, which cell types are primarily affected by serelaxin? Also, how does serelaxin impact the involvement of Epithelialmesenchymal transition (EMT) and endothelial–mesenchymal transition (EndMT)?

What are the results of immunostaining with fibrosis markers other than α-SMA?

Round 2

Reviewer 3 Report

The author has not conducted any additional experiments to address my concerns. I am too disappointed in the author.

Author Response

(The authors gave the same response as above.)
